# Early genetic evolution of driver mutations in uveal melanoma

James J. Dollar [1,2], Christina L. Decatur [1,2], Ezekiel Weis [3,4], Amy C. Schefler[5], Miguel A. Materin[6], Timothy S. Fuller[7], Alison H. Skalet[8,9], David A. Reichstein[10], Ivana K. Kim [11], Kisha D. Piggott[12], Hakan Demirci[13], Thomas A. Aaberg Jr.[14,15,16], Prithvi Mruthyunjaya[17], Basil K. Williams Jr.[18], Eugene Shildkrot [19], Scott C. N. Oliver [20], Devron H. Char[21], Antonio Capone Jr.[22], John O. Mason III[23], Scott D. Walter[24,25], Michael M. Altaweel[26], Jill R. Wells[27], Dan S. Gombos[28], Jay S. Duker[29], Peter G. Hovland[30], Tony Tsai [31], Cameron Javid[32], Michael A. Durante[1,2], Kyle R. Covington[33], Song Zhang [34,35], Zelia M. Correa[1,2] & J. William Harbour [35,36] ✉

Uveal melanoma (UM) is an aggressive eye cancer that frequently results in metastatic death despite successful primary tumor treatment. Subclinical micrometastasis is thought to occur early, when tumors are small and difficult to distinguish from benign nevi. However, the early genetic evolution of UM is poorly understood, and biomarkers for malignant transformation are lacking. Here, we perform integrated genetic profiling of 1140 primary UMs, including 131 small tumors. A clinically available 15-gene expression profile (15-GEP) prospectively validated by our group is more accurate than driver mutations for predicting patient survival. Small tumors are significantly more likely to be in earlier stages of genetic evolution than larger tumors. Further, the 15-GEP support vector machine discriminant score predicts small tumors undergoing transformation from low-risk Class 1 to high-risk Class 2 profile. These results shed light on the early genetic evolution of UM and move us closer to a molecular definition of malignant transformation in this cancer type.

Uveal melanoma (UM) is a deadly cancer of the eye with a high propensity for metastasis[1]. UM can be divided into four prognostically significant subtypes based on a 15-gene expression profile (15-GEP; Class 1 or Class 2) combined with the expression status of the cancer-testis antigen *PRAME* (negative or positive)[2–6]. This 15-GEP/*PRAME* classifier was recently validated in the Collaborative Ocular Oncology Group Study Number 2 (COOG2), a large international prospective multicenter biomarker study[6]. Within this molecular landscape, there are two clusters of highly recurrent UM-associated mutations (UMAMs)[7]. The first cluster consists of initiating mutations in one of four members of the G$_q$ signaling pathway (*GNAQ*, *GNA11*, *CYSLTR2*, and *PLCB4*)[8–12]. G$_q$ mutations do not appear to be sufficient for malignant transformation without the acquisition of further genomic aberrations, as they are also found in benign uveal nevi[8,9,13–15]. The second cluster comprises the BSE (*BAP1*, *SF3B1*, or *EIF1AX*) mutations, which are thought to signify malignant transformation and are associated with high, intermediate and low metastatic risk, respectively[16–19]. Mutations in *GNAQ*, *GNA11*, *CYSLTR2*, *PLCB4*, *SF3B1*, and *EIF1AX* are small somatic variants that are easily detected by next generation sequencing (NGS). In contrast, mutations in *BAP1* comprise a variety of deleterious alterations, some of which can be challenging to detect with NGS[20]. *BAP1* mutations are usually somatic but occasionally arise in the germline, and they become fully manifest by loss of the other allele by whole chromosome 3 loss[16,21]. BSE mutations and associated

copy number variations (CNVs) arise in the primary tumor around the same time during a punctuated evolutionary burst[14,20,22,23], although the timing of this event during genetic evolution remains unclear.

Since UM is thought to micrometastasize early, when tumors are small[24], thereby explaining the high metastatic rate despite successful primary tumor treatment[25], there remains a critical unmet need to elucidate the early genetic events in UM tumorigenesis and to better understand the molecular transition from benign nevus to malignant melanoma. Unfortunately, our current understanding of UM genetic evolution is inferred almost exclusively from large primary tumors that were treated by enucleation (eye removal)[14,20,22,23] or from metastatic tumors[26,27]. This lack of knowledge regarding small tumors is due in large part to their being treated by observation or eye-sparing therapies, where genetic analysis is limited to small biopsy samples.

Here, we characterize the mutation landscape, infer early genetic evolution, and evaluate the prognostic significance of UMAMs in a large multicenter prospective study of a real-world cohort of cases across the full spectrum of UM tumor size. We developed and analytically validated a targeted NGS panel for robust detection of all seven recurrent UMAMs using residual tumor biopsy material obtained during standard of care prognostic testing[28]. We find the prognostic value of 15-GEP and *PRAME* expression classification is superior to all UMAMs. Importantly, we identify that a low 15-GEP discriminant score predicts which UM are undergoing transformation from low-risk Class 1 to high-risk Class 2 expression profile. These findings expand our understanding of the early genetic evolution of UM and provide actionable insights for patient management.

## Results

### Patient cohort

Of 1687 subjects enrolled in COOG2, 1140 met inclusion criteria for this report, which included the presence of at least one UMAM (Supplementary Fig. 1). Baseline demographic and clinical information are summarized in Supplementary Table 1. Median age at study entry was 64.3 years (range, 18–99 years), including 550 (48.3%) female patients and 590 (51.8%) male patients. Baseline tumor thickness averaged at 5.5 mm (ranging 1.0–18.0 mm), while the mean tumor diameter was 12.6 mm (ranging 3.0–28.9 mm). Ciliary body involvement was present in 201 (17.6%) of the tumors. 15-GEP was Class 1 in 716 (62.8%) cases and Class 2 in 424 (37.2%) cases. *PRAME* expression was negative in 757 (66.4%) cases and positive in 383 (33.6%) cases. Median follow-up was 52.8 months. Metastatic disease was detected in 229 (20.1%) patients, and the median time to metastasis among patients with an event was 21.9 months (range, 17.3–79.9 months). Local tumor recurrence was identified in 54 (4.7%) patients with a median time of 28.5 months (range, 3.5–82.2 months) after biopsy/primary enucleation, with 28 (51.9%) of these patients subsequently developing metastatic disease.

### Landscape of uveal melanoma-associated mutations

UMAM NGS results are summarized in Fig. 1a–c; Supplementary Data 1 and Supplementary Fig. 2. $G_q$ mutations were detected in *GNAQ* in 558 (48.9%), *GNA11* in 530 (46.5%), *PLCB4* in 25 (2.2%), and *CYSLTR2* in 14 (1.2%) cases. BSE mutations were detected in *BAP1* in 364 (31.9%), *SF3B1* in 194 (17.0%), and *EIF1AX* in 304 (26.7%) cases. Associations between UMAMs and clinical and molecular features are summarized in Supplementary Data 2. *GNAQ* mutations were associated with Class 1 tumors ($p < 0.0001$), decreased patient age ($p = 0.008$), decreased tumor diameter ($p = 0.003$) and tumor thickness ($p = 0.0006$). Conversely, *GNA11* mutations were associated with Class 2 tumors ($p = 0.0005$), *PRAME*(+) status ($p = 0.05$), increased patient age ($p = 0.002$), increased tumor diameter ($p = 0.004$) and tumor thickness ($p = 0.002$), and they showed a near-significant association with ciliary body involvement ($p = 0.06$). $G_q$ mutations were mutually exclusive, except for 6 (0.5%) cases in which a $GNAQ^{Q209P}$, $GNA11^{Q209L}$,

$GNAQ^{R183Q}$ or $GNA11^{R183C}$ recurrent hotspot mutation was accompanied by a rare $GNAQ^{P193T}$, $GNAQ^{T175M}$, $CYSLTR2^{S154N}$ or $PLCB4^{D630N}$ mutation (Fig. 1d). *BAP1* mutations were associated with Class 2 tumors ($p < 0.0001$), *PRAME*(+) status ($p < 0.0001$), increased patient age ($p < 0.0001$), increased tumor diameter ($p < 0.0001$), increased tumor thickness ($p < 0.0001$), ciliary body involvement ($p < 0.0001$), mutations in *GNA11* ($p = 0.01$), *PLCB4* ($p = 0.02$) and *CYSLTR2* ($p = 0.02$), and absence of mutations in *GNAQ* ($p < 0.0001$). The spectrum of *BAP1* mutation types did not differ significantly between Class 1 and Class 2 tumors (Fig. 1c). *SF3B1* mutations were associated with Class 1 tumors ($p < 0.0001$), *PRAME*(+) status ($p < 0.0001$), decreased patient age ($p < 0.0001$), increased tumor diameter ($p = 0.02$), and brown iris color ($p = 0.008$). *EIF1AX* mutations were associated with Class 1 tumors ($p < 0.0001$), PRAME(-) status ($p < 0.0001$), decreased tumor diameter ($p < 0.0001$), lack of ciliary body involvement ($p < 0.0001$), and male sex ($p < 0.0001$). By and large, BSE mutations were mutually exclusive, with only 26 (2.3%) cases harboring two BSE mutations, including *BAP1* and *SF3B1* in 5 cases, *BAP1* and *EIF1AX* in 15 cases, and *SF3B1* and *EIF1AX* in 6 cases (Fig. 1d). No cases harbored all three BSE mutations.

### Prognostic significance of UMAMs

The prognostic significance of each UMAM was evaluated by Cox regression. In univariate analysis (Supplementary Table 2), *BAP1* was the only UMAM associated with shorter MFS (HR = 5.9, $p < 0.0001$), whereas *EIF1AX* (HR = 0.2, $p < 0.0001$), *SF3B1* (HR = 0.5, $p = 0.0009$), and *GNAQ* (HR = 0.8, $p = 0.05$) mutations were associated with longer MFS (Supplementary Fig. 3). *BAP1* (HR = 4.3, $p < 0.0001$) and *GNA11* (HR = 1.4, $p = 0.007$) mutations were associated with shorter OS, whereas *EIF1AX* (HR = 0.4, $p < 0.0001$), *SF3B1* (HR = 0.5, $p = 0.0005$), and *GNAQ* (HR = 0.7, $p = 0.001$) were associated with longer OS (Supplementary Fig. 4). In multivariate Cox analysis of MFS (Supplementary Table 3), when 15-GEP was entered into the model, mutations in *BAP1*, *EIF1AX*, *GNAQ*, and *GNA11* were rendered non-significant, and mutations in *SF3B1* became associated with shorter (rather than longer) MFS (HR = 1.7, $p = 0.03$). In the multivariate Cox analysis of OS, all UMAMs became non-significant when 15-GEP was entered into the model. Among Class 1 tumors, when *PRAME* status was entered into a multivariate Cox model, mutations in *SF3B1* became non-significant for MFS and OS. Thus, the combination of 15-GEP and *PRAME* renders all UMAMs non-significant and redundant for prognostic testing in UM.

### Insights into early genetic evolution from small tumors

To date, almost all genetic studies in UM have been performed on large enucleated tumors[20,22], but these represent only a small minority of the most advanced cases[1,29]. We hypothesized that smaller tumors, which are usually treated with eye-sparing therapies or observed for growth prior to treatment[30], may reveal insights into the early genetic evolution of UM. Thus, we compared 131 small tumors (defined as having thickness ≤ 2.5 mm and diameter ≤ 12 mm) based on thresholds established in previous reports using the 15-GEP[30,31] to the remaining 1009 larger tumors (Fig. 2a–c and Supplementary Data 3). Small tumors were more likely than larger tumors to be Class 1 ($p < 0.0001$), *PRAME*(−) ($p = 0.004$), $BAP1^{wt}$ ($p = 0.0006$), and to lack any BSE mutation ($p = 0.001$), suggesting that most or all UM begin as small Class 1 tumors that later acquire a BSE mutation during tumor growth. Average tumor purity was lower for small tumors (mean, 58.6% ± 3.1%) compared to larger tumors (mean, 81.9% ± 0.8%)(Wilcoxon test, $p < 0.0001$). Additionally, the discriminant score—the distance a given sample is from the 15-GEP support vector machine (SVM) decision boundary[32]—was significantly lower in small versus larger Class 2 tumors ($p = 0.003$)(Fig. 2d), potentially suggesting that small Class 2 tumors with low discriminant scores may have recently transitioned from small Class 1 tumors.

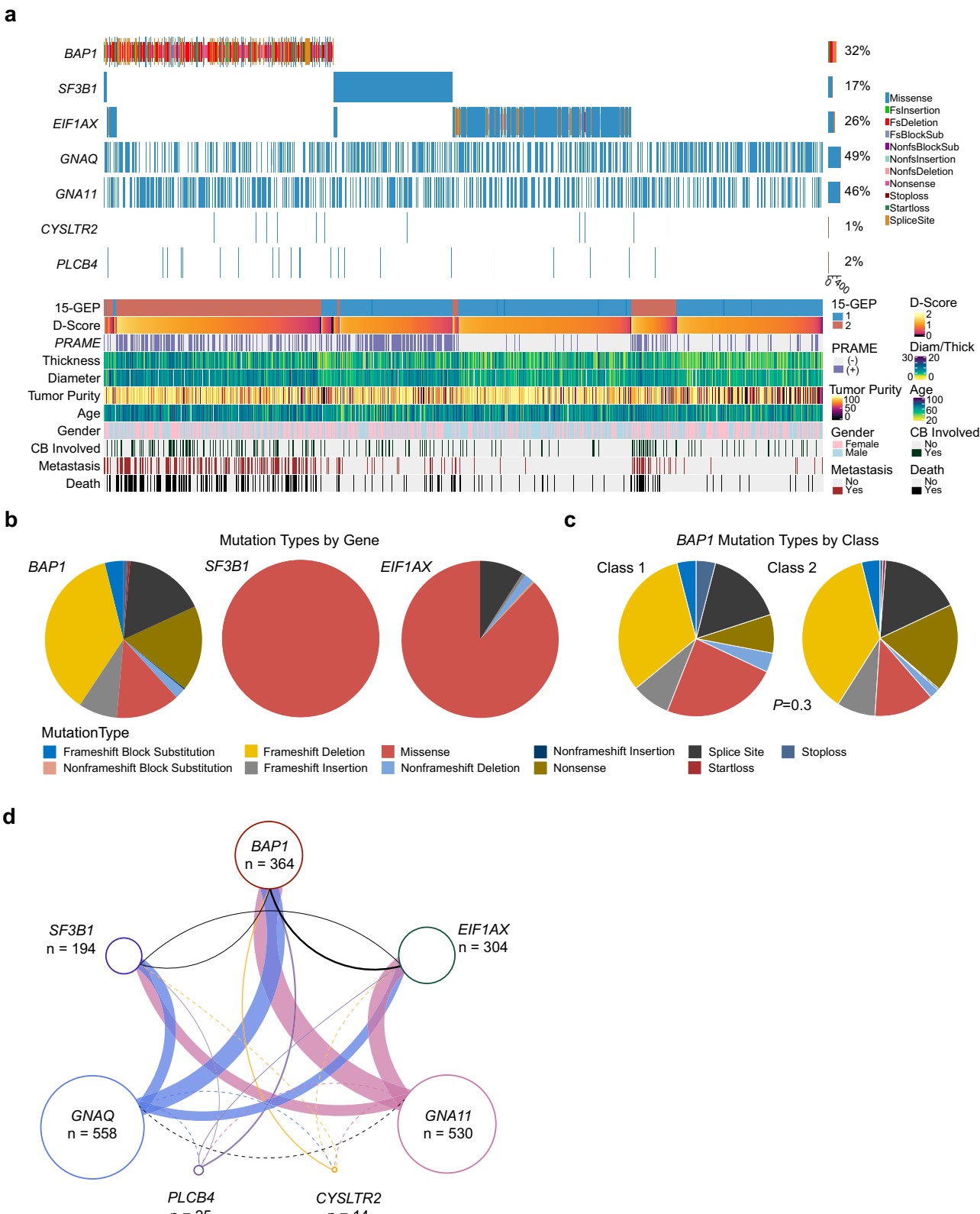

**Fig. 1 | Genetic landscape of uveal melanomas. a** Oncoprint of 1140 primary uveal melanomas, demonstrating the 7 canonical uveal melanoma associated mutations (UMAMs), along with 15-GEP status, *PRAME* status, tumor thickness (millimeters), tumor diameter (millimeters), gender, metastatic status (yes or no), and survival status (alive or deceased). Pie charts summarizing variant types for *BAP1*, *SF3B1*, and *EIF1AX* mutations **b** for all samples with at least one mutation (*n* = 836 tumors) and **c** for *BAP1* mutations in Class 1 (*n* = 25 tumors) and Class 2 tumors (*n* = 339 tumors). Significance was calculated by two-tailed Fisher's exact test. **d** Connectivity plot indicating co-occurring mutations, with connector color representing G$_q$ mutation (blue, *GNAQ*; mauve, *GNA11*; purple, *PLCB4*; yellow, *CYSLTR2*), and connector thickness corresponding to the number of cases. Dashed lines indicate ≤2 cases. Variant types described in the "Methods" section, and relevant data are provided in the Source data file. 15-GEP 15-gene expression profile, *PRAME*(+) *PRAME* positive, *PRAME* (−) *PRAME* negative, Diam tumor diameter, Thick tumor thickness, CB ciliary body, *D*-score 15-GEP support vector machine discriminant score.

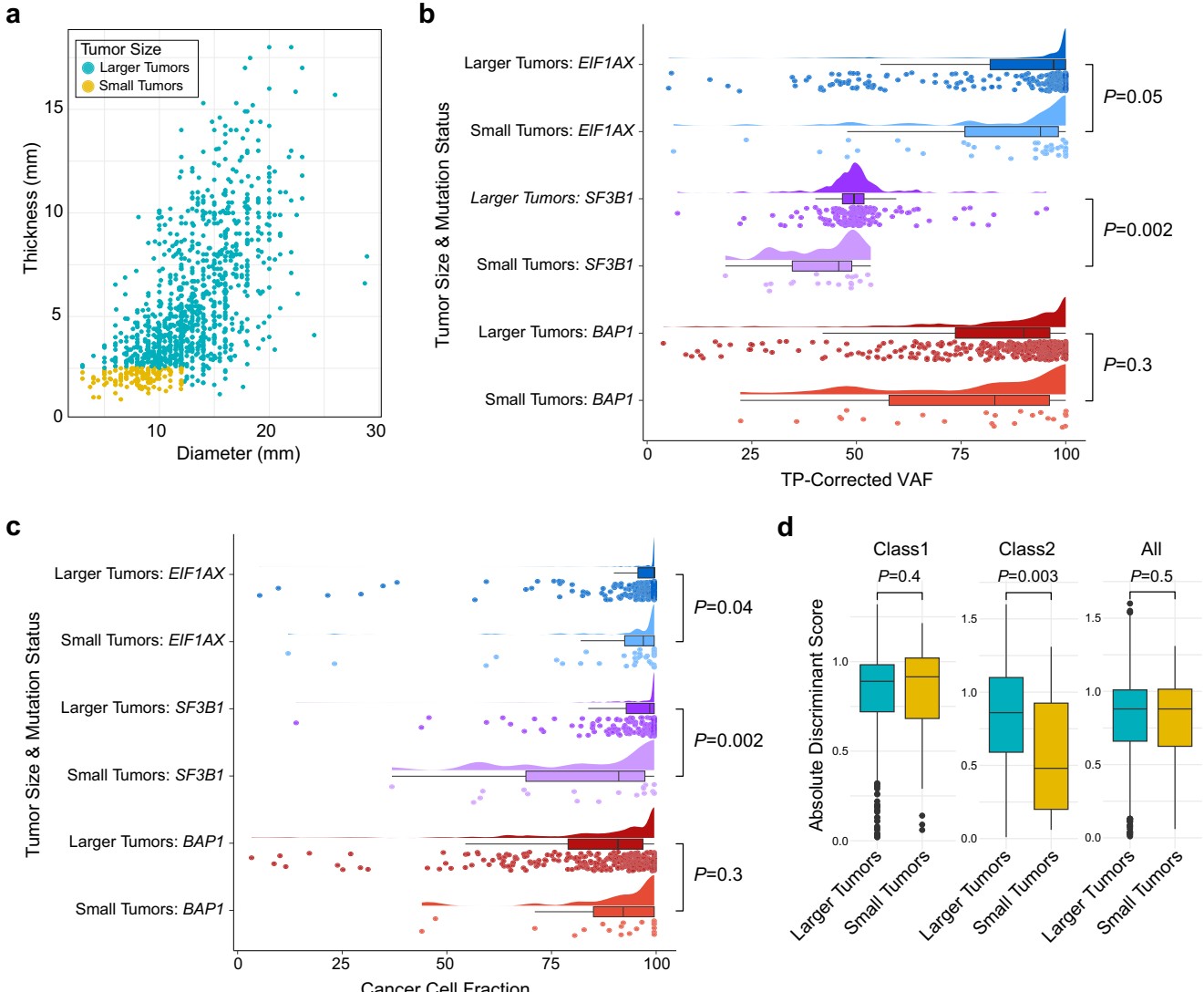

**Fig. 2 | Comparison of cancer cell fraction and 15-GEP discriminant score in small versus larger uveal melanomas. a** Scatter plot displaying the distribution of tumor thickness and diameter for 131 small tumors (yellow dots) versus 1009 larger tumors (green dots). **b** Raincloud plot of TP-corrected VAF for each BSE mutation in small ($n = 81$ tumors) versus larger tumors ($n = 761$ tumors). **c** Raincloud plot of cancer cell fraction (CCF) for each BSE mutation in small ($n = 76$ tumors) versus larger tumors ($n = 696$ tumors). **d** Box plot comparing the 15-GEP discriminant score for small tumors (yellow boxes) versus larger tumors (green boxes), comparing Class 1 ($n = 716$ tumors), Class 2 ($n = 424$ tumor) and all tumors ($n = 1140$ tumor). For box plots in (**b**–**d**), the box center line, lower boundary, and upper boundary display the median, 25th percentile, and 75th percentile, respectively. The distance between box boundaries reflects the interquartile range (IQR). Lower whiskers extend to the minima, or the lowest value up to 1.5 times the IQR from the lower box boundary. Upper whiskers extend to the maxima, or the highest value up to 1.5 times the IQR from the upper box boundary. Continuous variables were compared by two-tailed Wilcoxon rank-sum test. Relevant data are available within the Source data file. BSE, mutation in *BAP1*, *SF3B1*, or *EIF1AX*; $CCF_{BSE}$, cancer cell fraction for each BSE mutation, TP tumor purity, VAF variant allele frequency, mm millimeter.

## Insights into early genetic evolution from discordant tumors

While most cases exhibited the expected relationships between 15-GEP Class and *BAP1* status, there was a small subset of discordant cases (Fig. 3a–f), including 25 (3.5%) Class 1 tumors with a *BAP1* mutation (Fig. 4a, b and Supplementary Data 4). While *BAP1* mutation types did not differ significantly between Class 1 and Class 2 tumors (Fig. 1c), we further investigated potential functional differences in *BAP1* mutations between the two tumor classes using the recently described saturation genome editing (SGE) database for BAP1[33]. After excluding 106 complex *BAP1* mutations involving ≥5 nucleotide alterations, we successfully mapped 218 of the remaining 258 (>80%) *BAP1* mutations to the database. 97.7% of *BAP1* mutations (213/218) were functional classified as depleted, indicating a high concordance between our mutation-calling methodology and the SDE methodology (Supplementary Fig. 5a). Importantly, there was no significant difference in deleterious

categorization ($P = 0.3$) or functional scores ($p = 0.3$) for *BAP1* mutations in Class 1 versus Class 2 tumors (Supplementary Fig. 5a, b).

Class 1/*BAP1*[mut] tumors were associated with decreased tumor diameter ($p = 0.0006$), decreased tumor thickness ($P = 0.006$), and decreased discriminant score ($p < 0.0001$) compared to Class 1/*BAP1*[wt] tumors, suggesting that (1) most Class 1 tumors that acquire *BAP1* mutations do so when they are small and then convert to Class 2 before they have grown to a larger size, (2) *BAP1* mutations may be less likely to arise in Class 1 tumors above a certain size, possibly because the selective advantage has been satisfied by another aberration (e.g., *SF3B1* or *EIF1AX* mutation), and (3) the transition from Class 1 to Class 2 after acquiring a *BAP1* mutation is accompanied by a progressive decrease in the discriminant score on the Class 1 side of the decision boundary before increasing on the Class 2 side. Further, there was no difference in MFS or OS between Class 1/*BAP1*[mut] tumors compared to

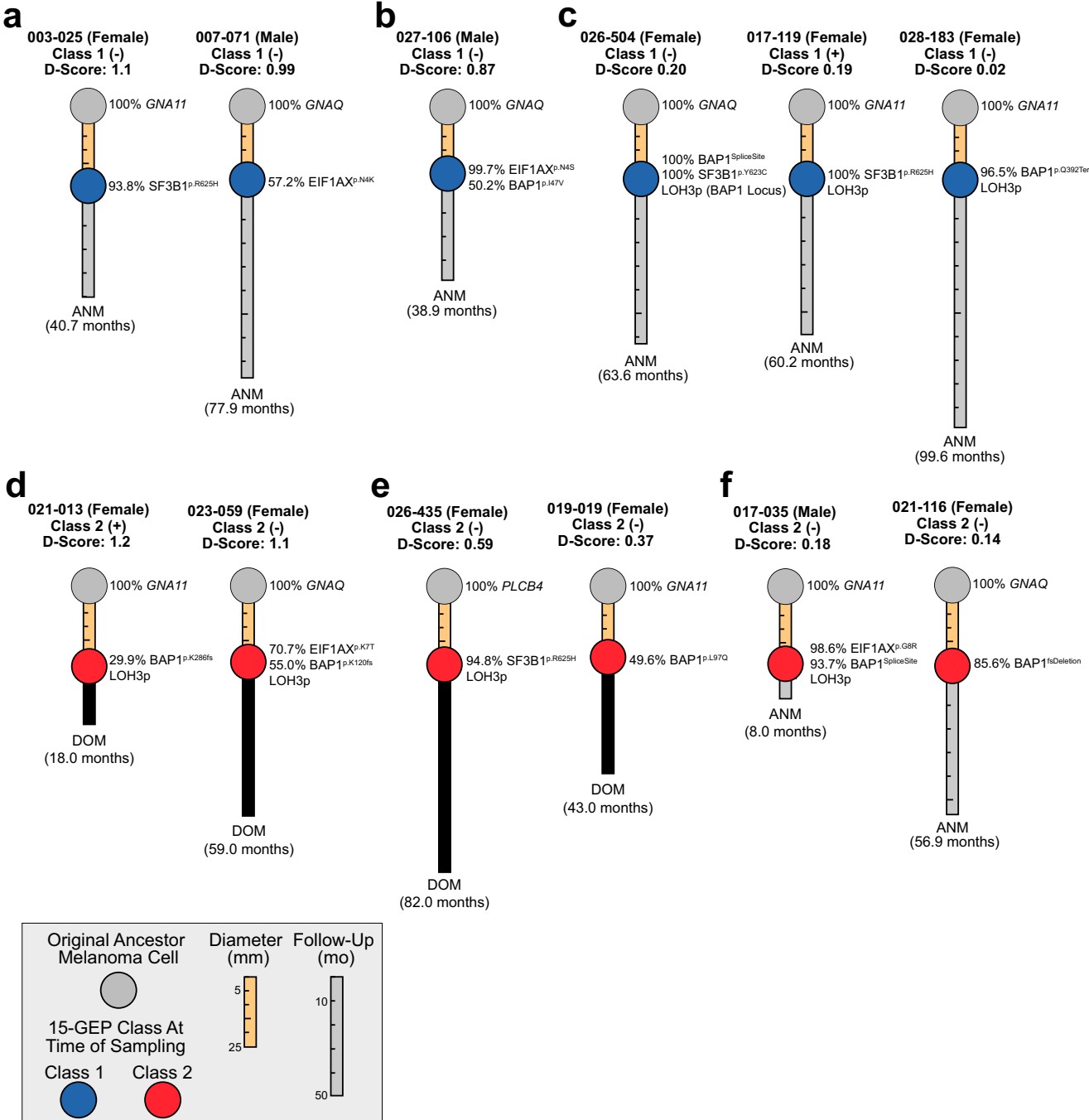

**Fig. 3 | Insights into early genetic evolution from 15-GEP discriminant score and BSE cancer cell fraction. a** Typical Class 1 tumors with high discriminant scores: Case #003-025, near-clonal *SF3B1* mutation; Case #007-071, sub-clonal *EIF1AX* mutation. **b** Discordant Class 1 tumors with high discriminant score: Case #027-106, near-clonal *EIF1AX* mutation and subclonal *BAP1* mutation. **c** Discordant Class 1 tumors with low discriminant scores: Case #026-504, clonal *BAP1* and *SF3B1* mutations and partial LOH3p involving a limited region around the *BAP1* locus; Case #017-119, clonal *SF3B1* mutation and LOH3p; Case #028-183, near-clonal BAP1 mutation, LOH3p and very low discriminant score (0.02). **d** Class 2 tumors with high discriminant scores and bi-allelic *BAP1* loss: Case #021-013, sub-clonal *BAP1* mutation; Case 023-059, near-clonal *EIF1AX* mutation and subclonal *BAP1* mutation. **e** Class 2 tumors with intermediate discriminant scores: Case #026-435, near-clonal *SF3B1* mutation and LOH3p but no detectable *BAP1* mutation; Case #019-019, sub-clonal *BAP1* mutation and no detectable LOH3p. **f** Class 2 tumors with low discriminant scores: Case #017-035, near-clonal *EIF1AX* and *BAP1* mutations with LOH3p; Case #021-116, near-clonal BAP1 mutation with no detectable LOH3p. The length of the connector between the ancestor melanoma cell (gray circle) and the melanoma (blue or red circle for Class 1 or Class 2, respectively) is proportional to tumor diameter (in millimeters) at the time of tumor sampling. The length of the extension beyond the melanoma is proportional to time to death or last follow-up (in months) with final status indicated. Survival status for living and dead patients is indicated by gray or black bar, respectively. Mutation nomenclature is described in the "Methods" section. Relevant data are available in the Source data file. 15-GEP 15-gene expression profile, ANM alive no metastasis, DOM dead of metastasis, *D*-score support vector machine discriminant score, LOH3p loss of heterozygosity of chromosome 3p; (−) *PRAME* negative, (+) *PRAME* positive.

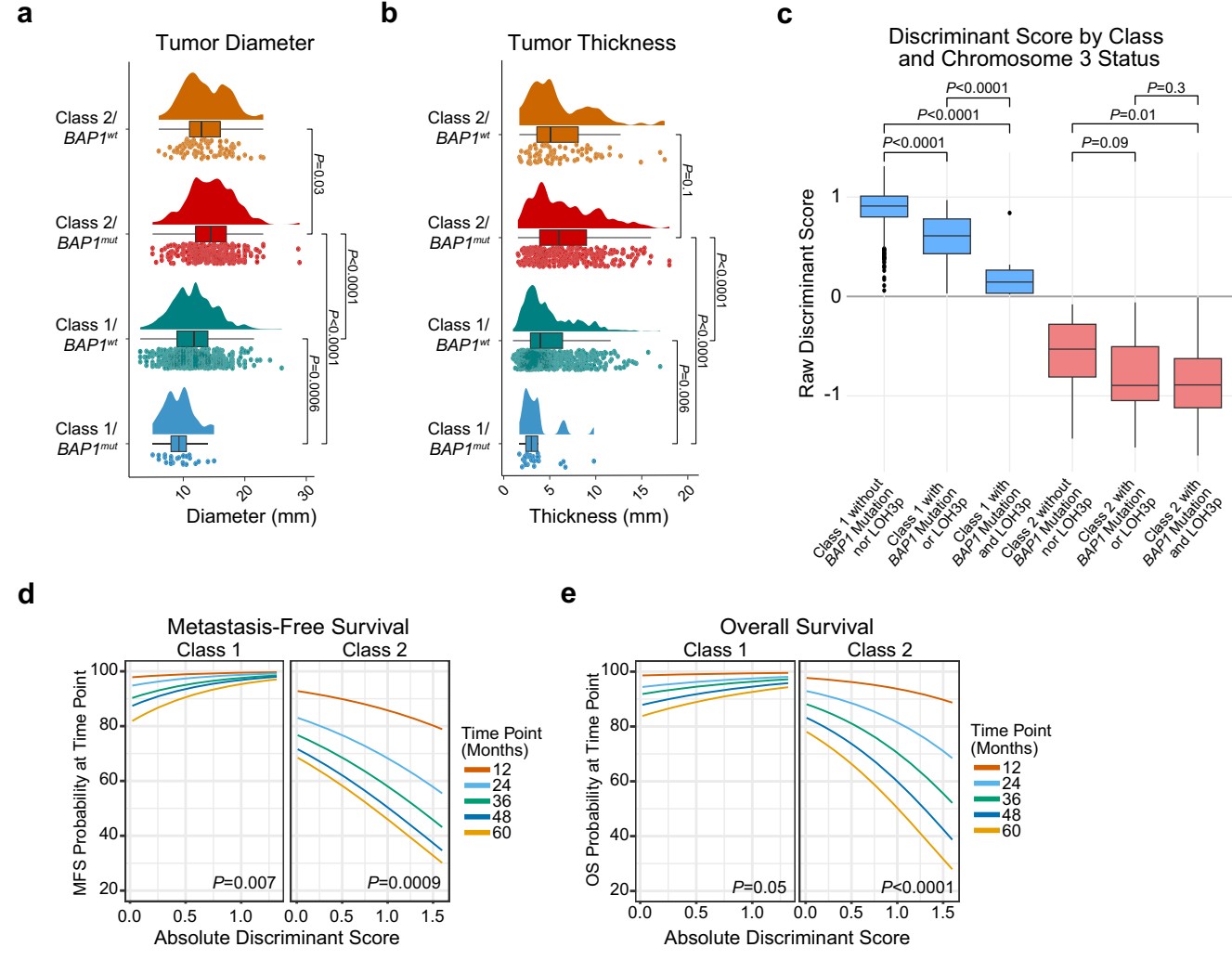

**Fig. 4 | Features of *BAP1* mutations by 15-GEP Class status. a** Raincloud plot depicting tumor diameter in relation to 15-GEP Class and *BAP1* mutation status (*n* = 1140 tumors). **b** Raincloud plot depicting tumor thickness in relation to 15-GEP Class and *BAP1* mutation status (*n* = 1140 tumors). **c** Box plot comparing raw discriminant scores by 15-GEP Class and *BAP1* allelic dosage reflected in *BAP1* mutation and LOH3p status (*n* = 905 tumors). Box center line, lower boundary, and upper boundary for box plots in (**a–c**) represent the median, first quartile, and third quartile, respectively, while box range reflect the interquartile range (IQR). Lower whiskers extend to the minima, or the lowest value up to 1.5 times the IQR from the first quartile. Upper whiskers extend to the maxima, or the highest value up to 1.5 times the IQR from the third quartile. Survival analysis plots displaying the

**d** metastasis-free survival and **e** overall survival probabilities for Class 1 (*n* = 715 tumors) and Class 2 (*n* = 418 tumors) UM according to absolute discriminant score at specified time points including 12 (red curves), 24 (light blue curves), 36 (green curves), 48 (dark blue curves), and 60 (orange curves) months. Significance for continuous variables was determined by two-tailed Wilcoxon rank-sum test. Significance for survival analysis was calculated by Cox proportional hazard analysis by Wald test. All data are available in the Source data file. Exact *p* values for thickness and diameter in Class 2/*BAP1*mut versus Class 1/*BAP1*wt were $4.8 \times 10^{-29}$ and $1.8 \times 10^{-16}$, respectively, and in Class 2/*BAP1*mut versus Class 1/*BAP1*mut were $2.6 \times 10^{-10}$ and $3.8 \times 10^{-7}$, respectively. 15-GEP 15-gene expression profile, MFS metastasis-free survival, OS overall survival, *BAP1*wt *BAP1* wildtype, *BAP1*mut *BAP1* mutant.

all Class 1/*BAP1*wt tumors (Supplementary Fig. 3 and 4), nor compared to a propensity score matched cohort of 75 Class 1/*BAP1*wt tumors (Supplementary Fig. 6a–c). The lack of survival difference could be explained by several factors: (1) since Class 1/*BAP1*mut tumors are generally small, any real decrease in survival may be very small and require longer follow-up to be detected, and (2) patients with Class 1/*BAP1*mut tumors may be among those most likely to be cured by effective local treatment by preventing early micrometastasis. Longer follow-up will be required to discern between these possibilities.

**Insights into early genetic evolution from discriminant score and cancer cell fraction**

Since the Class 2 signature results from bi-allelic loss of *BAP1*[16], we inferred the temporal relationship between *BAP1* loss and 15-GEP switch from Class 1 to Class 2 using the SVM discriminant score and cancer cell fraction (CCF) for *BAP1* (CCF$_{BAP1}$) in a subgroup of 905 cases

in which copy number status was available for the *BAP1* locus at chromosome 3p21. As anticipated, progressive decrease in BAP1 protein dosage (via mutational inactivation or chromosomal loss of the gene) was accompanied by a shift from Class 1 to Class 2 and an inversion of the discriminant score (Fig. 4c), with lower discriminant scores being associated with worse outcome in Class 1 tumors and better outcome in Class 2 tumors (Fig. 4d, e). We next evaluated CCFs for each BSE mutation. As expected, increasing CCF$_{SF3B1}$ and CCF$_{EIF1AX}$ were associated with larger tumor size (Supplementary Table 4), suggesting that these mutations usually arise in small tumors and progressively outcompete preexisting UM cells during tumor growth. Unexpectedly, however, there was no association between CCF$_{BAP1}$ and tumor size (Supplementary Table 4), nor was there an association between CCF$_{BAP1}$ and discriminant score (Spearman correlation, $R = -0.1$, $p = 0.07$), MFS or OS (Supplementary Table 5). Taken together, these findings suggest that BSE mutations usually occur early in

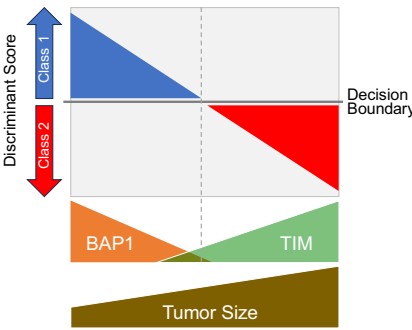

**Fig. 5 | Hypothesis for relationship between *BAP1* dosage, tumor immune microenvironment, and discriminant score.** *BAP1* dosage decreases as BAP1-deficient tumor cells outcompete BAP1-wildtype tumor cells, leading to altered composition of infiltrating immune cells in the tumor immune microenvironment (TIM). Since the 15-GEP includes genes expressed in tumor cells, immune cells or both, inversion of the SVM discriminant score from the Class 1 side to the Class 2 side of decision boundary occurs progressively as the transcriptional effects of *BAP1* loss accrue in both tumor and immune cells. This would explain why there is not a strict association between the fraction of cancer cells harboring mutant *BAP1* ($CCF_{BAP1}$) and the discriminant score, as the rate at which the TIM changes following BAP1 loss may differ between individuals. This would also explain why transitional tumors with low discriminant score tend to be small, whereas larger tumors, which have had longer for these transcriptional changes to occur, tend to have high discriminant scores.

the genetic evolution of UM when tumors are small. However, in the case of *BAP1*, mutational inactivation triggers a progressive transcriptomic shift from Class 1 to Class 2 accompanied by a decrease in discriminant score on the Class 1 side of the SVM decision boundary followed by an increase on the Class 2 side that is not tightly linked to $CCF_{BAP1}$ but likely also depends on alterations that *BAP1* loss causes to the tumor immune microenvironment (Fig. 5).

## Discussion

This study provides a comprehensive overview of the genetic landscape of UM across a real-world spectrum of tumor size, including small tumors previously excluded from most genetic analyses. Our findings suggest that the recurrent genomic aberrations that give rise to the archetypal evolutionary trajectories in UM[20] usually arise early when tumors are small. We confirmed the high, intermediate and low metastatic risk associated with *BAP1*, *SF3B1*, and *EIF1AX* mutations, respectively, but we found that these mutations are inferior to the 15-GEP/*PRAME* classifier for predicting metastasis-free and overall survival. Further, we found that the 15-GEP SVM discriminant score is a better indicator of tumors in transition from Class 1 to Class 2 than is the fraction of cancer cells harboring a *BAP1* mutation ($CCF_{BAP1}$). These findings shed light on the genetic evolution of UM and provide a clinically actionable framework for the precision treatment of selected small tumors at an earlier stage that may improve survival.

An unresolved question in the field is whether Class 2 tumors arise from Class 1 tumors or from a distinct precursor cell. We found that small tumors were much more likely to be Class 1 and to lack *BAP1* mutations compared to larger tumors. Further, we identified tumors that appeared to be in transition between Class 1 and Class 2 with low discriminant score and/or subclonal *BAP1* mutations (Fig. 3), consistent with the hypothesis that most or all Class 2 tumors arise from Class 1 tumors following bi-allelic loss of *BAP1*. We confirmed that BSE mutations are usually mutually exclusive, but there were some cases in which an *SF3B1* or *EIF1AX* mutation was followed by a *BAP1* mutation, indicating that the former does not absolutely protect from the latter and that BSE mutations may occasionally coexist and compete during early tumor evolution. Small tumors were also more likely to lack any

BSE mutation, suggesting that they were sampled early during genetic evolution and may not be fully transformed. The lower tumor purity in small lesions was likely due to an increased chance of aspirating surrounding normal cells and may have contributed to the lower detection rate of BSE mutations. However, since a Gq mutation was detected in all of these cases, a putative undetected BSE mutation would have necessarily been present at a low CCF, which is still consistent with our conclusion that some small tumors are still evolving their canonical UMAMs at the time of biopsy. It will be critical to determine how to distinguish between small tumors that can be safely monitored without treatment versus those that are likely to acquire high-risk genetic features if left untreated.

Among the seven recurrent UMAMs, only the BSE mutations exhibited independent prognostic significance, and even the BSE mutations were insignificant when the 15-GEP/*PRAME* classifier was included in multivariate analysis for both metastasis free and overall survival. The superiority of gene expression profiling over mutational analysis may have several explanations. First, it is likely that some Class 2/*BAP1*wt tumors actually had *BAP1* mutations that were undetectable with standard NGS methods[20]. Long read sequencing methods may improve the ability to detect such *BAP1* mutations[34]. Second, there was no significant correlation between $CCF_{BAP1}$ and survival, and there was no $CCF_{BAP1}$ threshold where the 15-GEP switched from Class 1 to Class 2, suggesting that there are additional factors beyond $CCF_{BAP1}$ that determine the Class 2 signature. Indeed, we previously showed that gene expression from tumor-infiltrating immune cells contributes substantially to the 15-GEP[23] and that *BAP1* loss in UM cells alters gene expression in adjacent immune cells[35]. Consequently, the 15-GEP appears to represent a functional snapshot of the transcriptional state of both cancer and immune cells in the tumor microenvironment that more accurately reflects metastatic propensity than does mutation analysis alone (Fig. 5).

While the $CCF_{BAP1}$ was of less prognostic value than anticipated, the 15-GEP SVM discriminant score provided unexpected insights into early tumor evolution and prognosis. We found that low discriminant score on either the Class 1 or Class 2 side of the decision boundary may indicate tumors in transition between these two states (Fig. 5). As such, a low discriminant score does not necessarily indicate low confidence but rather, it functions as a prognostic modifier associated with worse prognosis in Class 1 tumors and better prognosis in Class 2 tumors (Fig. 4d, e). A limitation of the chromosome copy number calling method is that it did not allow for precise CCF determination, such that LOH3p was assumed to be at ~100% CCF. However, this assumption is reasonable based on previous findings[14,20,22]. Further work is warranted to determine how best to incorporate the discriminant score into precision clinical management, perhaps by identifying small tumors in transition that should be treated promptly rather than observed.

It has been suggested that *GNA11* is a more potent oncogene than *GNAQ* mutations because *GNA11* mutations may be detected more frequently in metastatic tumors[10,36]. However, the present study does not support this claim. Whereas *GNA11* mutations were associated with some high-risk features, such as increased patient age and increased tumor size, they were not associated with MFS and only weakly associated with OS. Further, *GNA11* mutations were associated with *BAP1* mutation status and Class 2 status, but they were rendered non-significant for both MFS and OS when either *BAP1* mutation status or 15-GEP were entered into a multivariate Cox model. Thus, *GNA11* mutations are associated with other high-risk features but do not appear to have independent prognostic significance and do not appear to be more potent than GNAQ mutations. Of further interest regarding G$_q$ mutations, these were mutually exclusive as expected in all except six cases, which were of particular interest. In these six cases, a canonical G$_q$ hotspot mutation—*GNAQ*Q209P, *GNA11*Q209L, *GNAQ*R183Q or *GNA11*R183C—was accompanied by a rare G$_q$ pathway mutation—*GNAQ*P193T, *GNAQ*T175M, *CYSLTR2*S154N, or *PLCB4*D630N (Supplementary

Data 1). In 4 of these cases, the rare mutation was present at a higher variant allele frequency (VAF) than the hotspot mutation, suggesting that they occurred first but may have left residual selective pressure that led to the acquisition of another oncogenic $G_q$ mutation. If *GNA11* mutations were more potent than *GNAQ* mutations, we hypothesize that cases might be found in which a *GNAQ* mutation was followed by a *GNA11* mutation, but none were detected. While we did not find prognostic value for $G_q$ mutations independent of 15-GEP/*PRAME*, we did demonstrate the value of using the $G_q$ mutation VAF to estimate tumor purity (Supplementary Fig. 7a, b). A potential limitation of this method is the inability to detect whole genome doubling, which could potentially skew the VAF of heterozygous mutations. However, whole genome doubling is rare in uveal melanoma and limited to a small minority (<7%) of large, advanced cases[22]. Since our study comprised less than 10% of such advanced cases, this limitation is unlikely to have influenced our findings or conclusions.

In summary, this study confirms the prognostic value of UMAMs but demonstrates the inferiority of mutational analysis to the 15-GEP/*PRAME* classifier for prognostication. Nevertheless, UMAMs are relatively uncommon in other cancer types and can be useful for confirming the diagnosis of UM, which can be difficult in centers without specialized ocular cytopathology expertise[37]. The most unexpected finding was the value of the SVM discriminant score for inferring the evolutionary state of small tumors in transition between Class 1 and Class 2, which moves us closer to a quantitative molecular method for inferring the malignant potential of uveal melanocytic tumors that straddle the line between benign nevus and small melanoma—a subject of considerable controversy in the field[30,38,39]. These findings are timely in light of prevailing evidence suggesting that UMs may metastasize when they are small and difficult to distinguish from benign nevi[24,30,40,41], which could explain the failure of primary tumor treatment to prevent metastasis. Based on these results, a new prospective study is being planned to determine whether the discriminant score can be used in conjunction with the 15-GEP/*PRAME* classifier to guide the precision management of small uveal melanocytic tumors of indeterminate malignant potential by identifying lesions that are of sufficient risk of micrometastasis to warrant prompt treatment while sparing the vastly more abundant benign nevi that overlap in size[42]. Further studies and longer follow-up of this cohort will be important to further refine these prognostic tools for precision patient management.

## Methods

### Patient enrollment

This research complies with all relevant ethical regulations, and approval was obtained by the Federal Wide Assurance from the Office of Human Research Protections and Institutional Review Board (IRB) or Ethics Committee in accordance with policies at each participating center, with oversight by the University of Miami IRB. Participating IRBs included the Metro Health Institutional Review Board (Foundation for Vision Research); University of Wisconsin Health Sciences Institutional Review Board; Emory University Institutional Review Board; Western Institutional Review Board (for Associated Retinal Consultants, Tumori Foundation, Texas Retina Associates, Retina Associates of Arizona, Retina Consultants of Alabama, Tennessee Retina, and Retinal Consultants Medical Group); University of Cincinnati Institutional Review Board; University of Michigan Medical School Institutional Review Board; Tufts Health Sciences Campus Institutional Review Board; University of Texas MD Anderson Cancer Center Institutional Review Board; Catholic Health Initiatives Institute for Research and Innovation Institutional Review Board (Colorado Retina Associates); Massachusetts Eye and Ear Human Studies Committee; Duke University Health System Institutional Review Board for Clinical Investigations; Stanford University Institutional Review Board; Colorado Multiple Institutional Review Board; Washington University in St.

Louis Institutional Review Board; Houston Methodist Research Institute Institutional Review Board; University of Virginia Institutional Review Board for Health Sciences Research; Hartford HealthCare Institutional Review Board; Health Research Ethics Board of Alberta; Oregon Health & Science University Institutional Review Board; and University of Miami Institutional Review Board. Between January 2017 and April 2020, COOG2 enrolled 1687 subjects with UM involving the choroid, ciliary body and/or iris across 26 ocular oncology centers in the U.S. and Canada and prospectively monitored these subjects for metastatic progression and outcome. Informed written consent was obtained from each patient. Primary treatment was performed according to the standard at each center. Exclusion criteria included patient age less than 18 years, diagnosis of a uveal tumor other than UM (e.g., metastatic cancer), prior radiotherapy, inadequate sample for molecular analysis, and patient withdrawal from the study. Prior photodynamic therapy or transpupillary thermotherapy were allowed if there was evidence of tumor regrowth. No participants were excluded based on sex, ethnicity, or race, which was self-reported data. Gender was recorded from medical records and used as a proxy for biological sex in this study. No additional data on gender identity was collected. For this analysis, a data lock was performed on March 4, 2024. Subjects were not included for this report if they had a primary iris melanoma ($n = 101$ cases), lacked adequate residual biopsy material for successful sequencing ($n = 212$ cases) or had no detectable UMAM ($n = 234$ cases).

### Tumor sample analysis

All subjects underwent standard clinical genetic testing of the primary tumor prior to treatment using a commercial 15-GEP prognostic test (DecisionDx®-UM, Castle Biosciences, Inc., Friendswood, TX, USA) and qPCR assay for *PRAME* mRNA expression (DecisionDx®-PRAME, Castle Biosciences, Inc., Friendswood, TX, USA). This testing was performed in a CAP-accredited, CLIA-certified clinical laboratory, as previously described[4,43]. DecisionDx®-UM employs SVM to assign each sample to Class 1 (low metastatic risk) or Class 2 (high metastatic risk), and it assigns a discriminant score as a measure of confidence based on the distance of a given sample to the SVM decision boundary[32]. DecisionDx®-PRAME renders a result of positive or negative based on a validated threshold[44].

Approximately ~25% of each clinical sample was retained for analysis on a UMAM NGS panel (DecisionDx®-UMSeq, Castle Biosciences, Inc.), as previously described[28]. Variants were sequenced with Ion GeneStudio S5 Prime Sequencer (ThermoFisher Scientific, Waltham, MA, USA) and processed with Ion Reporter (Version 5.6) software. Variant detection, analysis, and annotation was conducted with Ion Torrent Suite Browser (Version 5.8) and Ion Reporter using human reference sequence hg19. Sequencing quality assessment was conducted for each run, including total yield, useable reads, percent polyclonal reads, and amplicon coverage, as previously described[28]. Sample-specific sequencing quality metrics are included in Supplementary Data 1.

Mutations were classified as nonsense (introduction of a premature stop codon), stop-loss or start-loss (loss of stop or start codon preventing translation), frameshift insertion or deletion (shift of codon reading frame via addition or subtraction of a non-triplet set of nucleotides), non-frameshift insertion or deletion (addition or removal of a codon without shifting the reading frame), block substitution (alteration of multiple sequential codons), splice site alteration (alteration of splice donor or acceptor site), and missense (substitution of one amino acid). All of the following variants were called pathogenic: nonsense, stop-loss, start-loss, frameshift and non-frameshift insertions and deletions, and block substitutions. Splice site alterations were called pathogenic if predicted to result in splice acceptor or donor site loss or gain variant as predicted by a SpliceAI (Version 1.3) score greater than or equal to 0.5[45]. Missense variants were called

pathogenic if they: (1) were previously reported as pathogenic in the ClinVar Database[46], (2) exhibited a SIFT (Version 5.2.2) score less than or equal to 0.05, or (3) exhibited a PolyPhen2 (Version 2.2.2) score greater than or equal to 0.5. All genetic variants that were called pathogenic were classified as tier I, II, or III according to the guidelines of the College of American Pathologists (CAP), American Society of Clinical Oncology, and Association for Molecular Pathology[47].

### Functional assessment of *BAP1* mutations using saturation genome editing database

*BAP1* mutations involving complex alterations (≥5 nucleotide changes) were excluded from analysis and the remainder were converted from hg19 to hg38 reference genomes using the Broad Institute *Liftover* tool (https://liftover.broadinstitute.org/) (Version 03-03-2024). Mutations were mapped to a CRISPR-based SGE database for *BAP1*, matching mutations based on hg38 start position, reference allele(s), and mutant allele(s) to retrieve the previously reported SGE functional classifications and scores[33]. Significance of functional classification was determined by two-tailed Fisher's exact test, and significance of functional scores was determined by two-tailed Wilcoxon signed-rank test.

### Calculation of tumor purity, variant allele frequency, and cancer cell fraction

Tumor purity (TP), the percentage of cells in a sample that are tumor cells, was inferred from the VAF of the $G_q$ mutation, assuming that the $G_q$ mutation is the founder mutation, is a heterozygous alteration, and is therefore present at 50% VAF in tumor cells. In rare cases with more than one $G_q$ mutation, the mutation with the highest frequency (and presumably the earlier initiating mutation) was used. As such, $TP = min([VAF_{Gq\text{-mutant}} \times 2], 100\%)$. To validate the estimation of tumor purity based on VAF of $G_q$ mutations, we compared tumor purity estimation using VAF of $G_q$ mutation to that using chromosome CNVs in the UM TCGA cohort[22] using ABSOLUTE and FACETS. Statistical significance was determined using Pearson correlation (Supplementary Fig. 7).

The VAF for *BAP1*, *SF3B1*, and *EIF1AX* mutations was corrected for TP using the following equation: $TP\text{-corrected } VAF_{BSE} = VAF_{BSE}/TP$. Samples without a detectable $G_q$ mutation could not be corrected for VAF and, thus, were not included in analyses requiring TP-corrected $VAF_{BSE}$. Next, we estimated the CCF for each BSE mutation, representing the proportion of UM cells that harbor a given mutation, which requires a correction for allelic copy number. *SF3B1* is located on chromosome 2, which is not frequently altered in UM[20,22]. Thus, *SF3B1* mutations were assumed to be heterozygous and $CCF_{SF3B1} = min(TP\text{-corrected } VAF_{SF3B1} \times 2, 100\%)$. *EIF1AX* is located on the X chromosome, which is also rarely lost in UM[22]. Thus, gender was used to calculate mutant $CCF_{EIF1AX}$, where females were assumed to have an *EIF1AX* mutation at 50% and males at 100% of TP-corrected VAF. Thus, the $CCF_{EIF1AX}$ for females was calculated as $CCF_{EIF1AX} = min(TP\text{-corrected } VAF_{EIF1AX} \times 2, 100\%)$, whereas the $CCF_{EIF1AX}$ for males was assumed to be equal to TP-corrected $VAF_{EIF1AX}$. *BAP1* is located at chromosome 3p21[48], which frequently undergoes copy number loss in UM[20,22]. Thus, to detect loss of heterozygosity (LOH) and calculate CCF for *BAP1*, we developed a custom targeted CNV sequencing panel containing 74 loci across chromosome 3p that was performed on the same sample used for the 15-GEP/*PRAME* classifier and UMAM NGS panel. For *BAP1*-mutant tumors with retention of heterozygosity for chromosome 3p, the $CCF_{BAP1}$ was calculated as $CCF_{BAP1} = min(TP\text{-corrected } VAF_{BAP1} \times 2, 100\%)$. For tumors demonstrating LOH for chromosome 3p (LOH3p), $CCF_{BAP1}$ was assumed to be equal to TP-corrected $VAF_{BAP1}$.

For the custom CNV sequencing panel, B-allele frequencies and log fold-change (lfc) read depths across chromosome 3p were compared to a reference DNA panel of normals, comprising peripheral blood mononuclear cell samples from 64 patients. Variant call format (VCF) files were analyzed using Wheeljack (https://github.com/ covingto/KRCGTK/releases/tag/v0.1) (Version 0.1). Copy-number loss for chromosome 3p was detected by consistent b-allele frequencies at 100% and a decreased lfc read depth of less than 0. Isodisomy for chromosome 3p was identified by consistent b-allele frequencies at 100% and a lfc read depth of approximately 0. For downstream analyses, samples demonstrating either copy number loss or isodisomy for chromosome 3p were called as LOH3p, whereas samples without these aberrations were called as retention of heterozygosity for 3p. Calls were made by hand and adjudicated by 3 of the authors (J.J.D., C.L.D., and K.R.C.). Variability across b-allele and read depth plots was used to assign confidence scores with 0, 1, 2, and 3 corresponding to very low, low, medium, and high confidence, respectively. A confidence score of 2 or 3 was required for use in downstream analyses.

### Data management

REDCap (https://projectredcap.org/), a secure HIPAA-compliant application[49], was used for electronic data management, as previously described[6]. Baseline data included date of enrollment, date and method of biopsy, cytology result (if available), date and method of primary tumor treatment, patient age at study entry, sex, self-reported race and ethnicity, iris color (blue/green, intermediate, or brown), tumor diameter, tumor thickness, ciliary body involvement, and metastatic status. The American Joint Committee on Cancer (AJCC) 8th edition[50] was used for tumor staging. Follow-up data included local tumor recurrence (tumor regrowth in the eye or orbit following radiotherapy or in the orbit following enucleation), metastatic status, date and location of initial metastasis, systemic status at last follow-up, and date and cause of death. Molecular test results were entered into REDCap by Castle Biosciences, which was masked to other REDCap data. Each center was masked to data entered by other centers and by Castle Biosciences. Only the coordinating center and COOG2 Data Committee had access to all data.

Baseline and follow-up ophthalmic visits were performed as per standard of care at each center but typically included a comprehensive ophthalmic examination, fundus photography, optical coherence tomography, and ultrasonography performed at least every 3–4 months for the first year after treatment, every 4–6 months for the second year, and every 6–12 months thereafter. Baseline systemic imaging was typically performed with CT of the chest, abdomen, and pelvis. Subsequent systemic surveillance typically included imaging of the liver with CT, MRI or ultrasound at least twice a year, along with chest CT or chest X-ray at least once a year.

### Statistical analysis

Statistical analysis was performed using SAS 9.4 (SAS Institute, Cary, NC) and R (v4.2.2). Chi-square test was used to compare categorical variables unless expected frequencies were less than 5 for at least 25% of category cells, in which case Fisher exact test was used. Two-tailed Wilcoxon signed-rank test was used for comparing continuous variables. Statistical analysis of patient demographics and tumor characteristics for association with UMAMs compared all patients with a given mutation to all those without the mutation. All statistical tests were two-sided, and statistical significance was defined as $P < 0.05$. Differences in metastasis-free survival (MFS, time from primary tumor treatment to initial detection of metastatic disease) and overall survival (OS, time from primary tumor treatment to death from any cause) associated with a given factor were evaluated using Kaplan–Meier (KM) survival curves and the log-rank test. Propensity scores were calculated based on tumor thickness and tumor diameter to compare MFS and OS in Class 1 tumors that were wildtype versus mutant for *BAP1*, using a 3:1 matching ratio. Cox regression was used to assess the contribution of multiple factors influencing metastatic risk. Univariable and multivariable Cox models were constructed to assess the impact of variables both separately and in combination. Survival analysis for continuous variables (e.g., discriminant score) was performed

by calculating survival probabilities at specified time points using a time-to-event model that includes the continuous variable[51]. The sample size for the overall COOG2 study was determine as previously described[6]. The current study included all cases with complete genetic annotations available ($n = 1140$), with the most stringent comparison being the prognostic accuracy of 15-GEP versus BAP1 mutation status. Given that there were 110 cases with discordant genetic annotations (25 Class 1/BAP1[mut] and 85 Class 2/BAP1[wt] case), and assuming a 5-year MFS of approximately 90% for Class 1/BAP1[mut] and 50% for Class 2/BAP1[wt], we have ~80% power to detect a ~20% difference between the two discordant groups at 5% two-sided type I error.

## Reporting summary

Further information on research design is available in the Nature Portfolio Reporting Summary linked to this article.

## Data availability

Raw sequencing data generated for this study have been deposited in the Sequence Read Archive (SRA) database and the Genotypes and Phenotypes (dbGaP) Database under accession number phs004040.v1.p1 [http://www.ncbi.nlm.nih.gov/projects/gap/cgi-bin/study.cgi?study_id=phs004040.v1.p1]. Access to the data requires an approved application through dbGaP due to patient privacy concerns. Corresponding author can be contacted and will give permission if investigator requesting the data submits reasonable research application for raw data access, an agreement for non-commercial research use only, and the requested length of time for data access. Response to request will be made within 14 days after review of request. Data will be accessible for the requested length of time proposed if request is approved. The detailed cohort data (including mutation metrics, survival outcomes, tumor features, and patient details) analyzed in this study are available in Supplementary Information and at the Dryad Research Data Repository [https://doi.org/10.5061/dryad.z8w9ghxqk]. Forced call VCF files used for assessing *BAP1* heterozygosity also deposited at the same Dryad Research Data Repository. For TCGA UM cohort analysis, tumor purity data were accessed from Supplementary Table provided by Robertson et al.[22], while whole-exome sequencing analysis results were accessed from the Supplementary Data published with Field et al.[20]. *BAP1* SGE functional scores and classification data from Waters et al., 2023 are available with the Supplementary Data provided with the publication[33] and at https://github.com/team113sanger/Waters_BAP1_SGE. All data presented in main and Supplementary Figs. are available in the Source data file. Source data are provided with this paper.

## Code availability

Custom code used for calculations, including tumor purity and cancer cell fraction, as well as for visualization, is available at https://github.com/jwharbour/COOG2tools under the MIT License. All code is open source and freely available without restriction. The repository includes scripts for preprocessing, statistical analysis, and generating figures. This code includes use of publicly available R packages and software; full citations and version details are provided within the GitHub repository documentation. For inquiries or assistance, please contact the corresponding author. A permanent version of this code is accessible via Zenodo repository [https://doi.org/10.5281/zenodo.17298698][52].

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

## Acknowledgements

This research was supported by National Eye Institute to J.W.H. (R01 CA125970); Cancer Prevention and Research Institute of Texas Recruit-ment of Established Investigator Award to J.W.H. (RR220010); Research to Prevent Blindness, Inc. Senior Scientific Investigator Award to J.W.H.; National Cancer Institute Support Grants to Simmons Comprehensive Cancer Center, University of Texas Southwestern Medical Center to J.W.H. (P30 CA142543) and Sylvester Comprehensive Cancer Center, University of Miami to Z.M.C. (P30 CA240139); National Institute of General Medical Sciences Grant to the Medical Scientist Training Pro-gram of University of Miami for support of J.J.D. (T32 GM145462). National Eye Institute Core Grants to the Department of Ophthalmology, University of Texas Southwestern Medical Center to J.W.H. (P30 EY030413), Bascom Palmer Eye Institute and Department of Ophthal-mology, University of Miami Department of Ophthalmology to Z.M.C. (P30 EY014801) and Casey Eye Institute, Oregon Health & Science Uni-versity to A.H.S. (P30 EY010572); Research to Prevent Blindness, Inc. Challenge Grant to the Department of Ophthalmology, University of Texas Southwestern Medical Center to J.W.H.; Research to Prevent Blindness, Inc. Unrestricted Grants to Bascom Palmer Eye Institute, Department of Ophthalmology, University of Miami (Z.M.C.) and Casey Eye Institute, Oregon Health & Science University (A.H.S.); Malcolm M. Marquis, MD Endowed Fund for Innovation to Casey Eye Institute, Ore-gon Health & Science University to A.H.S.; and Castle Biosciences, Inc. grant to University of Miami to Z.M.C. The authors thank our patients who generously participated in this project. We acknowledge Katherina M. Alsina, PhD, Jason H. Rogers, MS, Jennifer J. Siegel, PhD, Jeff Wilkinson, PhD, Lauren Sholl, MS, and Daniel Vargas, MS, from Castle Biosciences, Inc., for data management support, and the following individuals for their dedication to accurate data entry at each study site: Angie Adler, Mutaz Al-Nawaflh, Corrina Azarcon, Buse Guneri Beser, Karina Bostwick, Nury Cabrera, Teja Chemudupati, Caroline Craven, Jessica Fitch, Nancy Gee, Ashley Go, Caleb Hartley, Mustafa Hashmi, Tyler Hendrickson, Gary Lamoureux, Anne Marie Lane, Kiley Lazarek, Ashton Leone, Ronan McCarthy, Audra Miller, Trece Mayhan, Monica Oxenreiter, Barbara Perez, Dayana Pineda, Nicki Plocharsky, Mary Preston, Kourtney Storey, Laurie Tavernier, Bonnie Verges, Holly Vincent, Brooke Waller, and Aaron Yeung.

## Author contributions

J.W.H. conceived of, designed, and acquired the financial support for the study. J.W.H., C.L.D., and Z.M.C. provided administrative support. E.W., A.C.S., M.A.M., T.S.F., A.H.S., D.A.R., I.K.K., K.D.P., H.D., T.A.A, P.M., B.K.W., E.S., S.C.N.O, D.H.C., A.C., J.O.M., S.D.W., M.M.A., J.R.W., D.S.G., J.S.D., P.G.H., T.T., C.J., Z.M.C., and J.W.H. provided study materials and

enrolled patients for the study. J.J.D., C.L.D., and P.M. collected and assembled data. J.J.D., K.R.C., M.A.D., S.Z., E.W., and J.W.H. conducted data analysis and interpretation. J.J.D. and J.W.H. wrote the original draft of the manuscript. All authors provided critical input and contributed to discussions and revisions of the manuscript.

## Competing interests

J.J.D., C.L.D., E.W., M.A.M., T.S.F., A.H.S., D.A.R., I.K.K., K.D.P., H.N., T.A.A., P.M., B.K.W., E.S., S.C.N.O., J.R.W., D.S.G., J.O.M., S.D.W., T.T., Z.M.C., and J.W.H. have acted as consultants for Castle Biosciences. K.R.C. is employed by Castle Biosciences. J.W.H. has received royalties for intellectual property related to prognostic testing in uveal melanoma that was licensed to Castle Biosciences. The remaining authors declare no competing interests. Castle Biosciences played no role in the conceptualization, design, data analysis, decision to publish, or preparation of the manuscript; they only contributed to data collection by depositing deidentified genetic data into an encrypted REDCap database without access to patient data.

## Additional information

[1]Department of Ophthalmology and Bascom Palmer Eye Institute, University of Miami Miller School of Medicine, Miami, FL, USA. [2]Sylvester Comprehensive Cancer Center, University of Miami Miller School of Medicine, Miami, FL, USA. [3]Department of Ophthalmology, Faculty of Medicine and Dentistry, University of Alberta, Edmonton, AB, Canada. [4]Division of Ophthalmology, Department of Surgery, Faculty of Medicine, University of Calgary, Calgary, AB, Canada. [5]Retina Consultants of Texas, Houston, TX, USA. [6]Department of Ophthalmology, Duke University, Durham, NC, USA. [7]Texas Retina Associates, Dallas, TX, USA. [8]Department of Ophthalmology and Casey Eye Institute, Oregon Health and Science University, Portland, OR, USA. [9]Knight Cancer Institute, Oregon Health and Science University, Portland, OR, USA. [10]Tennessee Retina, Nashville, TN, USA. [11]Department of Ophthalmology and Massachusetts Eye and Ear Infirmary, Harvard Medical School, Boston, MA, USA. [12]Department of Ophthalmology and Visual Sciences, Washington University, St. Louis, MO, USA. [13]Department of Ophthalmology and Visual Sciences, Kellogg Eye Center, University of Michigan, Ann Arbor, MI, USA. [14]Retina Specialists of Michigan, Grand Rapids, MI, USA. [15]Foundation for Vision Research, Grand Rapids, MI, USA. [16]Michigan State University College of Human Medicine, Grand Rapids, MI, USA. [17]Department of Ophthalmology and Byers Eye Institute, Stanford University, Stanford, CA, USA. [18]Department of Ophthalmology, University of Cincinnati, Cincinnati, OH, USA. [19]Department of Ophthalmology, University of Virginia, Charlottesville, VA, USA. [20]Department of Ophthalmology and Sue Anschutz-Rodgers Eye Center, University of Colorado, Aurora, CO, USA. [21]Tumori Foundation, San Francisco, CA, USA. [22]Associated Retina Consultants, Royal Oak, MI, USA. [23]Department of Ophthalmology, University of Alabama, Birmingham, AL, USA. [24]Retina Consultants, Hartford, CT, USA. [25]Helen and Harry Gray Cancer Center, Hartford, CT, USA. [26]Department of Ophthalmology, University of Wisconsin, Madison, WI, USA. [27]Department of Ophthalmology, Emory University, Atlanta, GA, USA. [28]Section of Ophthalmology, Department of Head and Neck Surgery, The University of Texas, MD Anderson Cancer Center, Houston, TX, USA. [29]Department of Ophthalmology and New England Eye Center, Tufts University, Boston, MA, USA. [30]Colorado Retina Associates, Denver, CO, USA. [31]Retinal Consultants Medical Group, Sacramento, CA, USA. [32]Retina Associates, Tucson, AZ, USA. [33]Castle Biosciences, Friendswood, TX, USA. [34]O'Donnell School of Public Health, University of Texas Southwestern Medical Center, Dallas, TX, USA. [35]Simmons Comprehensive Cancer Center, University of Texas Southwestern Medical Center, Dallas, TX, USA. [36]Department of Ophthalmology, University of Texas Southwestern Medical Center, Dallas, TX, USA. ✉e-mail: william.harbour@utsouthwestern.edu

