## [Transparent Peer Review file · Nature Communications]

Early Genetic Evolution of Driver Mutations in Uveal Melanoma

Corresponding Author: Professor J. William Harbour

Version 0:

Reviewer comments:

Reviewer #1

(Remarks to the Author)

The authors propose a more comprehensive analysis of uveal melanoma genetic landscape by examining small tumors, thereby overcoming the limitations of current knowledge regarding UM pathogenetics, which is primarily based on large tumors. Current insights into the genetics and pathogenesis of UM, through integrated multi-platform analyses, reveal a clear division of this disease into distinct groups concerning the risk of metastatic disease. However, some tumors classified as low-risk do metastasize, and current markers do not adequately predict the propensity for metastasis in these UMs. It is still unclear whether genetic and/or epigenetic alterations in a subpopulation of tumor cells can trigger an evolution from a relatively indolent state to an aggressive UM. Additionally, the tumor evolution model, which involves successive molecular alterations conferring a growth advantage that accumulates over time, has recently been challenged by the punctuated equilibrium or "big bang" model. This alternative model predicts an initial phase of genomic instability, followed by the emergence of stabilized clones carrying molecular lesions that confer a selective growth advantage. This work aims to define the etiology of UM and elucidate its pathobiology. This approach will foster a deeper understanding of the molecular mechanisms behind uveal melanoma, improving risk stratification, refining therapies, and boosting patient survival. Therefore, we decided to approve its publication with minor revisions

Minor Revisions:

Line 93-95 median follow-up data appear to be incorrectly linked to Supplementary Table 1.

Line 97-98 the % related to the sentence "with 28 (51.9%) of these patients subsequently developing metastatic disease" in which supplementary table is it listed? Not in supplementary Table 1

Line 105 I suggest to add Figure 1a together with Supplementary Table 3

Line 116 PLCB4 reported an error in the p-value, in the Supplementary Table 3 is indicated 0.02, in the text $P=0.01$

Line 134 it is not clear whether the information on MFS and OS shorter and longer is shown in Supp. Table 2 or 3, perhaps a supplementary figure with Kaplan Meyer curve is more impactful.

Line 134 in Supp. Table 5 it is not clear why BAP1, GNAQ and GNA11 line 13 and 14 of the table are shown twice.

Line 268-270 the sentence "If GNA11 mutations were more potent than GNAQ mutations, we might have expected to find cases where a GNAQ mutation was followed by a GNA11 mutation, but none were detected", seems to be an hypothesis that need more investigation studies and not a general conclusion.

Line 334 insert this acronym "variant allele frequency (VAF)" in line 266 where you first mentioned VAF

Line 418 the indicated link is not working

Line 737 I think the authors meant to say BAP1-wildtype tumor cells and not calls

The figure 1b and 1d are not mentioned in the text.

Figures:

Figure 1A improve the quality of the figure

Figure 1D indicate PLCB4 and CYSLTR2 in the figure

Figure 1 in general make the labeling more readable

In the legend of Figure 1 the description of Figure 1C is related to Figure 1D and the description of Figure 1C is missing

Line 739 I would remove the words Supervector Machine and leave only SVM from Figure 5

(Remarks on code availability)

Reviewer #2

(Remarks to the Author)

Dollar et al report on 1140 primary UMs from COOG2, using targeted NGS panel from tumour biopsies to understand primary UM's evolutionary dynamics. In particular, they characterise small tumours and find that many lack a BSE (BAP1, SETD2 or EIF1AX) mutation, suggesting that BSE mutations can be acquired later with tumour growth. To date, most analyses have been done on resected UMs, by definition more advanced stage- the difference in this cohort is the inclusion of primary UMs that were not surgically resected but only biopsied. They were profiled with a bespoke driver gene panel and for 15-gene expression and PRAME signatures previously described.

Previously known prognostic associations (for example with BAP1) were confirmed. With a multi-variate analysis, they demonstrate the gene expression signature to be superior prognosticator for MFS and OS when compared to secondary driver mutations in BAP1, SF3B1 and EIF1AX. They compared 131 small tumours to >1000 larger tumours finding that smaller tumours were less evolved. The gene expression signature showed likely recent switch from class 1 to 2 in some instances.

Comments/queries:

- Lines 105-109: not clear on the direction of these associations- both GNAQ and GNA11 are associated with increased tumour diameter for example?
- Lines 123-125 where >1 BSE mutation occurred, is there evidence that they are both pathogenic?
- Lines 157-172 show some unexpected associations with BAP1 mutations- could the authors categorise these mutations functionally including with the published saturation genome editing scores for BAP1?
- How were the size criteria for small tumours (<2.5mm thickness and <12mm diameter) selected? They do not fit with TNM staging (i.e. do not conform to T1 subsets (e.g. T1a)). For ease of clinical translation/applicability, could the analysis still hold if it included only T1a or all T1 tumours?
- How do these molecular predictors compare with clinical predictors of survival e.g. T staging, ciliary body involvement etc? It would be of interest to compare this, to further demonstrate clinical utility. Could it be possible to construct a combined clinical and molecular survival model?
- Within class 1 tumours, BAP1mt are smaller than BAP1wt, but no difference in survival outcomes. The authors state 'These findings suggest that there is a lag time following loss of BAP1 before the 15-GEP switches to Class 2 and the metastatic rate increases'. Can the authors please elaborate on how they arrived at this conclusion?
- CNAs are also prognostic in UM. Are data available for the other common CNAs in UM, apart from monosomy3? A 'custom CNV sequencing panel' is mentioned in the methods. It would be of interest to understand how the common CNAs correlate with small tumours and evolution, and to understand how class 2 / PRAME correlate with CNAs such as chr8 alterations, 16q loss and 1p loss (per Lalonde et al). What if these CNAs are added into the survival model? If they are also rendered insignificant by GEP/PRAME, that would be of interest, perhaps suggesting they all relate to the same signal.
- Where small tumours did not have a BSE mutation – can the authors confirm that there was enough tumour purity in the sample to call mutations?

(Remarks on code availability)

Reviewer #3

(Remarks to the Author)

In Dollar et al., the authors utilized targeted sequencing of driver genes in uveal melanomas collected from 1140 patients across multiple institutions over approximately a three year period. The authors identified driver mutations across uveal melanomas of varying diameter and thickness, and assessed the statistical relationships of these genetic events with matching clinical and demographic data. The authors also found that a discriminant score based on gene expression of 15 genes was a strong indicator of progression risk, and that adding PRAME RNA expression in the classifier performed well in predicting overall and metastasis free survival.

Overall this study represents an impressive sample set with a large volume of genetic data, and the results are also significant for determining the evolution of uveal melanoma which remains a highly aggressive tumor type.

Nonetheless, I have several critiques below on the analyses and interpretations of the sequencing data that underpin the study. I am particularly concerned regarding assessments of tumor purity, especially since these estimates are critical for the results that follow, as well as the study's novelty.

Major

My topmost concern regards tumor purity, which potentially impacts many of the downstream results. In this manuscript, the authors calculated tumor purity using the variant allele frequency of the Gq mutation in each case, assuming that each one is present in 100% of cancer cells and is also heterozygous. While I think these assumptions are reasonable, tumor purity is typically inferred using copy number alterations. I acknowledge that the authors chose the best option given their data, and their results may remain robust with additional analyses. Nonetheless, there are sequencing and other technical issues that could impact the Gq variant allele frequencies and thus may skew the inferred tumor purities. The authors should validate their method, perhaps by analyzing an independent dataset. For example, tumor purities in TCGA whole exome data were

typically inferred using copy number alterations. To validate whether Gq variant allele frequencies derive similar tumor purities as compared to somatic copy number alterations, the authors could analyze TCGA uveal melanoma data to correlate the Gq variant allele frequencies reported there with matching tumor purities inferred by TCGA copy number results. Further, tumor purities should be included as a row in the oncoprint in Figure 1a. The authors also need to consider whole genome doubling, as there are likely cases in their cohort impacted by this, as also found in a fraction of cases in TCGA (see Figure 1A in PMID 28810145). As an illustration of how critical the accuracy of tumor purities potentially are to this study, in Supplementary Table 2 I found that the small tumor subset selected by the authors had ~30% of the samples (40/131 tumors) with a tumor purity <25%, whereas the remaining tumors had ~6% of the samples (58/990 with tumor purity values) with a tumor purity of <25%. Why would small tumors have lower purities? This is important because the authors subsequently find that the smaller tumors were more likely to lack a BSE mutation, although in the context of lower purities it is possible that the BSE mutations were simply missed as false negatives. Overall, it is unclear how accurate the tumor purities are in this study, and what impact these inferences may have for the results.

My second concern is novelty. As the authors state, this sample set is one of the largest in uveal melanoma thus far. Yet the sample set itself is the subject of a previous publication, and almost all the findings here have been previously published, mostly by the authors themselves. I think the authors need to more convincingly delineate the novelty of this particular study, specifically which results are independent in this manuscript versus those that are simply extensions or validations of prior work.

Here are my intermediate concerns:

Were the tumor samples collected prior to the treatments listed in Supplementary Table 2? Were these samples FFPE, frozen, or a mix?

About 17% of tumor samples (n=290) failed sequencing, with the majority (n=234) having no UM driver mutation detected. Why? Providing some quality metrics of the sequencing results would help interpret this result.

What is the evidence in this manuscript specifically that BAP1 loss causes changes to the tumor immune microenvironment that impact the transcriptomic shift from Class 1 to Class 2? This seems like a result from a prior publication focusing on cell line experiments, not genomic data of human tissues. The discussion on immune cells seems unrelated to the data here, yet it is a major part of Figure 5?

The data availability is scant and the sequencing files should be provided in some form to a repository like dbGAP or EGA. The authors also need to provide sequencing coverage and mutant reads for each mutation so that readers can adequately assess the mutation calls.

Minor concerns:

What was the approach for defining "small" as having thickness less than or equal to 2.5 mm and diameter less than or equal to 12 mm? It seems arbitrary based on the plot in Figure 2a.

Figure 3 includes metastasis arrowheads in the legend but I do not see any on the timelines.

The authors mention "punctuated" evolution in the abstract, which is a separate topic of their prior work. What is the evidence in this manuscript for "punctuated" evolution? All of the tumors here could have followed a step-wise paradigm before the sample was collected (even the "small" tumors were millions to billions of cells by the time of collection). For example, an inferred founder with multiple drivers is consistent with "punctuated" evolution, but it is also consistent with step-wise evolution.

(Remarks on code availability)

Reviewer #4

(Remarks to the Author)

(Remarks on code availability)

Version 1:

Reviewer comments:

Reviewer #1

(Remarks to the Author)

The authors have replied to the comments resolving all the revisions.

(Remarks on code availability)

Reviewer #3

(Remarks to the Author)

The authors have comprehensively addressed my previous comments. I also agree the results in the new Supplementary Figure 7 (correlation of FACETS and Gq tumor purities) support the study and the rigor of the authors' approach.

(Remarks on code availability)

Reviewer #4

(Remarks to the Author)

(Remarks on code availability)

NCOMMS-25-08109 "Early Genetic Evolution of Driver Mutations in Uveal Melanoma"
Dollar et al.
April 7, 2025

REVIEWER COMMENTS

Reviewer expertise:

Reviewer #1: Uveal melanoma genomics and clinical research

Reviewer #2: Cancer genomics and evolution

Reviewer #3: Cancer genomics and evolution, uveal melanoma

Reviewer #4: Early-Career Researcher co-reviewer

Reviewer #1 (Remarks to the Author):

The authors propose a more comprehensive analysis of uveal melanoma genetic landscape by examining small tumors, thereby overcoming the limitations of current knowledge regarding UM pathogenetics, which is primarily based on large tumors. Current insights into the genetics and pathogenesis of UM, through integrated multi-platform analyses, reveal a clear division of this disease into distinct groups concerning the risk of metastatic disease. However, some tumors classified as low-risk do metastasize, and current markers do not adequately predict the propensity for metastasis in these UMs. It is still unclear whether genetic and/or epigenetic alterations in a subpopulation of tumor cells can trigger an evolution from a relatively indolent state to an aggressive UM. Additionally, the tumor evolution model, which involves successive molecular alterations conferring a growth advantage that accumulates over time, has recently been challenged by the punctuated equilibrium or "big bang" model. This alternative model predicts an initial phase of genomic instability, followed by the emergence of stabilized clones carrying molecular lesions that confer a selective growth advantage. This work aims to define the etiology of UM and elucidate its pathobiology. This approach will foster a deeper understanding of the molecular mechanisms behind uveal melanoma, improving risk stratification, refining therapies, and boosting patient survival.

Therefore, we decided to approve its publication with minor revisions

Minor Revisions:

Line 93-95 median follow-up data appear to be incorrectly linked to Supplementary Table 1.

RESPONSE: The median follow-up data in the text has been linked to the updated Supplementary Table 1.

Line 97-98 the % related to the sentence "with 28 (51.9%) of these patients subsequently developing metastatic disease" in which supplementary table is it listed?
Not in supplementary Table 1

RESPONSE: We have now added this to Supplementary Table 1.

Line 105 I suggest to add Figure 1a together with Supplementary Table 3

RESPONSE: Most of the clinical variables in Supplementary Table 3 are already included in the oncoprint in Figure 1a. We have now added age and ciliary body involvement, which were the other prognostically relevant clinical variables, along with discriminant score and tumor purity.

Line 116 PLCB4 reported an error in the p-value, in the Supplementary Table 3 is indicated 0.02, in the text $P=0.01$

RESPONSE: Thank you for catching this. It has been corrected in the updated manuscript.

Line 134 it is not clear whether the information on MFS and OS shorter and longer is shown in Supp. Table 2 or 3, perhaps a supplementary figure with Kaplan Meyer curve is more impactful.

RESPONSE: Kaplan-Meier curves for MFS and OS are included in Supplementary Figures 3 and 4.

Line 134 in Supp. Table 5 it is not clear why BAP1, GNAQ and GNA11 line 13 and 14 of the table are shown twice.

RESPONSE: These mutations are shown twice because they are included in two separate multivariate survival analysis models to elucidate whether GNAQ or GNA11 mutations were associated with independent prognostic significance when BAP1 mutations were included in the model. The answer was “no” – when BAP1 mutations were included in the model, neither GNAQ or GNA11 mutations provided independent significance, suggesting that the univariate analysis results showing an association of GNA11 mutations with worse prognosis (and GNAQ mutations with better prognosis) are most likely due to GNA11 mutations being associated with BAP1 mutations and GNAQ mutations being inversely associated with BAP1 mutations. This is explained in the text in lines 274-283.

Line 268-270 the sentence “If GNA11 mutations were more potent than GNAQ mutations, we might have expected to find cases where a GNAQ mutation was followed by a GNA11 mutation, but none were detected”, seems to be an hypothesis that need more investigation studies and not a general conclusion.

RESPONSE: We agree that this statement was meant as a hypothesis, not a conclusion. We have re-worded it as follows: "If *GNA11* mutations were more potent than *GNAQ* mutations, we hypothesize that cases might be found in which a *GNAQ* mutation was followed by a *GNA11* mutation, but none were detected."

Line 334 insert this acronym "variant allele frequency (VAF)" in line 266 where you first mentioned VAF

RESPONSE: The manuscript has now been adjusted accordingly.

Line 418 the indicated link is not working

RESPONSE: The link in the manuscript will be made publicly available upon manuscript acceptance. Please use this link to access a temporary page for peer review: <https://gitfront.io/r/jjdollar/5oEsj38fuFCj/COOG2tools/> Additionally, all software provided in the mentioned manuscript link is the same as the compressed software file provided for review as well.

Line 737 I think the authors meant to say BAP1-wildtype tumor cells and not calls

RESPONSE: Yes, this has been corrected.

The figure 1b and 1d are not mentioned in the text.

RESPONSE: Thank you for pointing this out. Figures 1b and 1d are now called out in the text (lines 101 and 113, respectively).

Figures:

Figure 1A improve the quality of the figure

RESPONSE: The final version for publication is high resolution, and this can be provided to the reviewer as needed.

Figure 1D indicate PLCB4 and CYSLTR2 in the figure

RESPONSE: PLCB4 and CYSLTR2 labels were cut off in the document sent to reviewers but is included in the final figure file (see screenshot below):

d

Figure 1 in general make the labeling more readable

RESPONSE: The final version of Figure 1 for publication is high resolution and much more readable. In addition, we have increased the font size where feasible.

In the legend of Figure 1 the description of Figure 1C is related to Figure 1D and the description of Figure 1C is missing

RESPONSE: This has been corrected.

Line 739 I would remove the words Supervector Machine and leave only SVM from Figure 5

RESPONSE: This has been corrected.

Reviewer #2 (Remarks to the Author):

Dollar et al report on 1140 primary UMs from COOG2, using targeted NGS panel from tumour biopsies to understand primary UM's evolutionary dynamics. In particular, they characterise small tumours and find that many lack a BSE (BAP1, SETD2 or EIF1AX) mutation, suggesting that BSE mutations can be acquired later with tumour growth. To date, most analyses have been done on resected UMs, by definition more advanced stage- the difference in this cohort is the inclusion of primary UMs that were not surgically resected but only biopsied. They were profiled with a bespoke driver gene panel and for 15-gene expression and PRAME signatures previously described.

Previously known prognostic associations (for example with BAP1) were confirmed. With a multi-variate analysis, they demonstrate the gene expression signature to be superior prognosticator for MFS and OS when compared to secondary driver mutations in BAP1, SF3B1 and EIF1AX. They compared 131 small tumours to >1000 larger tumours finding that smaller tumours were less evolved. The gene expression signature showed likely recent switch from class 1 to 2 in some instances.

Comments/queries:

- Lines 105-109: not clear on the direction of these associations- both GNAQ and GNA11 are associated with increased tumour diameter for example?

RESPONSE: Thank you catching this error. In all cases, we indicate the direction of association (e.g., increased or decreased), but we errantly indicated GNAQ mutations were associated with increased tumor diameter and thickness, when in fact GNAQ mutations were associated with decreased tumor diameter and thickness. This has been corrected.

- Lines 123-125 where >1 BSE mutation occurred, is there evidence that they are both pathogenic?

RESPONSE: Yes. We initially filtered for variants classified as tier I, II, or III according to the guidelines of the College of American Pathologists (CAP), American Society of Clinical Oncology (ASCO), and Association for Molecular Pathology (AMP). As such, the following variants were called pathogenic: nonsense, stop-loss, start-loss, frameshift and non-frameshift insertions and deletions, block substitutions, and splice site alterations. Splice site alterations were called pathogenic if predicted to result in splice acceptor or donor site loss or gain variant as predicted by a SpliceAI score greater than or equal to 0.5 (PMID: 30661751). Missense variants were called pathogenic if they: (1) were previously reported as pathogenic in the ClinVar Database, (2) exhibited a SIFT score less than or equal to 0.05, or (3) exhibited a PolyPhen2 score greater than or equal to 0.5. We expanded our description of this methodology in the Methods section (lines 356-371) and included calculated SIFT and PolyPhen scores in Supplementary Table 1.

- Lines 157-172 show some unexpected associations with BAP1 mutations- could the authors categorise these mutations functionally including with the published saturation genome editing scores for BAP1?

RESPONSE: This is an excellent idea and have undertaken the suggested analysis. Keeping in mind that the saturation genome editing method published for BAP1 (PMC11250367) was intended primarily for evaluation of germline BAP1 mutations, it was expected that some of the more complex somatic BAP1 mutations that can be found in tumors but not in the germline would not be evaluable. Indeed, 106 of the 364

BAP1 mutations were not evaluable because they involved complex alterations involving ≥ 5 nucleotides and were predominantly frameshift mutations with high confidence of their deleterious effect on BAP1 function. Additionally, another 40 mutations were not identified in the SGE database. These were primarily frameshift mutations (33/40, 82.5%), strongly suggesting these mutations are also deleterious. We were successful in evaluating the remaining 218 BAP1 mutations using the SGE database. These included 17 Class 1 and 201 Class 2 tumors. The vast majority of the BAP1 variants that we called pathogenic (213/218, 97.7%) were classified as "depleted" in the SGE database, indicating deleterious impact on protein function. Only four mutations were classified as "unchanged" (from one Class 1 tumor and three Class 2 tumors), and surprisingly, a single mutation from a Class 2 tumor was classified as "enriched."

Fisher Exact Test showed that "depleted" BAP1 mutations were not significantly enriched in either 15-GEP Class 1 or Class 2 UM (Fisher's $P = 0.3$). Furthermore, SGE functional scores showed no significant difference between Class 1 ($n = 17$) and Class 2 UM ($n = 201$) (Wilcoxon test, $P = 0.3$). These findings support the notion that BAP1 mutations in Class 1 tumors are not enriched for non-pathogenic variants, and that they are as likely as those in Class 2 tumors to be pathogenic. We have now incorporated this analysis in lines 165-173 and in Supplementary Figure 5 as additional support for findings.

- How were the size criteria for small tumours (<2.5mm thickness and <12mm diameter) selected? They do not fit with TNM staging (i.e. do not conform to T1 subsets (e.g. T1a)). For ease of clinical translation/applicability, could the analysis still hold if it included only T1a or all T1 tumours?

RESPONSE: The <2.5mm thickness and <12mm diameter criteria were chosen based on objective studies using the 15-GEP as a standardized external biomarker to identify thresholds between small uveal melanocytic lesions at low versus high risk of malignant

transformation. A thickness threshold of >2.25 mm was found to be optimal for distinguishing small choroidal melanocytic lesions at low risk for malignant transformation versus those at high risk (PMC6291343). Similarly, a diameter threshold of 12 mm was found to distinguish between low and high risk in small choroidal melanocytic tumors (PMC4966166; PMID: 26596399). In contrast, the size cutoffs for the 8th edition AJCC were somewhat arbitrary and not widely accepted. Further, the system as a whole has not been found to be as accurate as the 15-GEP/PRAME classifier (PMC6214741; PMC11421563). Thus, we would prefer to maintain our current criteria, which were empirically derived and more widely accepted. We have added this an explanation in the Method section (lines 148-150).

- How do these molecular predictors compare with clinical predictors of survival e.g. T staging, ciliary body involvement etc? It would be of interest to compare this, to further demonstrate clinical utility. Could it be possible to construct a combined clinical and molecular survival model?

RESPONSE: We addressed this in the first publication of the COOG2 study, in which the 15-GEP/PRAME classifier was shown to be superior to T staging and to render all individual clinical predictors redundant/nonsignificant, except for a small independent contribution by tumor diameter (PMC11421563). In the present 2nd report of the COOG2 study, we show that uveal melanoma mutations are also prognostically redundant/nonsignificant if 15-GEP/PRAME are included in the model; they provide no additional prognostic accuracy (though they are valuable for other reasons as described in the manuscript). Thus, we found no value in a combined clinical and molecular survival model.

- Within class 1 tumours, BAP1mt are smaller than BAP1wt, but no difference in survival outcomes. The authors state 'These findings suggest that there is a lag time following loss of BAP1 before the 15-GEP switches to Class 2 and the metastatic rate increases'. Can the authors please elaborate on how they arrived at this conclusion?

RESPONSE: We have re-written this section to improve clarity (lines 174 – 190).

- CNAs are also prognostic in UM. Are data available for the other common CNAs in UM, apart from monosomy3? A 'custom CNV sequencing panel' is mentioned in the methods. It would be of interest to understand how the common CNAs correlate with small tumours and evolution, and to understand how class 2 / PRAME correlate with CNAs such as chr8 alterations, 16q loss and 1p loss (per Lalonde et al). What if these CNAs are added into the survival model? If they are also rendered insignificant by GEP/PRAME, that would be of interest, perhaps suggesting they all relate to the same signal.

RESPONSE: We are developing a CNV panel that looks at all of chromosome 3 (the

data in this study only include the 3p region around BAP1), as well as chromosomes 6 and 8. However, that work is still under development and beyond the scope of this study.

- Where small tumours did not have a BSE mutation – can the authors confirm that there was enough tumour purity in the sample to call mutations?

RESPONSE: Please see detail response to a similar question by Reviewer #3.

Reviewer #3 (Remarks to the Author):

In Dollar et al., the authors utilized targeted sequencing of driver genes in uveal melanomas collected from 1140 patients across multiple institutions over approximately a three year period. The authors identified driver mutations across uveal melanomas of varying diameter and thickness, and assessed the statistical relationships of these genetic events with matching clinical and demographic data. The authors also found that a discriminant score based on gene expression of 15 genes was a strong indicator of progression risk, and that adding PRAME RNA expression in the classifier performed well in predicting overall and metastasis free survival.

Overall this study represents an impressive sample set with a large volume of genetic data, and the results are also significant for determining the evolution of uveal melanoma which remains a highly aggressive tumor type.

Nonetheless, I have several critiques below on the analyses and interpretations of the sequencing data that underpin the study. I am particularly concerned regarding assessments of tumor purity, especially since these estimates are critical for the results that follow, as well as the study's novelty.

Major

My topmost concern regards tumor purity, which potentially impacts many of the downstream results. In this manuscript, the authors calculated tumor purity using the variant allele frequency of the Gq mutation in each case, assuming that each one is present in 100% of cancer cells and is also heterozygous. While I think these assumptions are reasonable, tumor purity is typically inferred using copy number alterations. I acknowledge that the authors chose the best option given their data, and their results may remain robust with additional analyses. Nonetheless, there are sequencing and other technical issues that could impact the Gq variant allele frequencies and thus may skew the inferred tumor purities. The authors should validate their method, perhaps by analyzing an independent dataset. For example, tumor

purities in TCGA whole exome data were typically inferred using copy number alterations. To validate whether G_q variant allele frequencies derive similar tumor purities as compared to somatic copy number alterations, the authors could analyze TCGA uveal melanoma data to correlate the G_q variant allele frequencies reported there with matching tumor purities inferred by TCGA copy number results. Further, tumor purities should be included as a row in the oncoprint in Figure 1a. The authors also need to consider whole genome doubling, as there are likely cases in their cohort impacted by this, as also found in a fraction of cases in TCGA (see Figure 1A in PMID 28810145). As an illustration of how critical the accuracy of tumor purities potentially are to this study, in Supplementary Table 2 I found that the small tumor subset selected by the authors had ~30% of the samples (40/131 tumors) with a tumor purity <25%, whereas the remaining tumors had ~6% of the samples (58/990 with tumor purity values) with a tumor purity of <25%. Why would small tumors have lower purities? This is important because the authors subsequently find that the smaller tumors were more likely to lack a BSE mutation, although in the context of lower purities it is possible that the BSE mutations were simply missed as false negatives. Overall, it is unclear how accurate the tumor purities are in this study, and what impact these inferences may have for the results.

RESPONSE: We agree that using the G_q mutation as the basis for calculating tumor purity was our best option, and in fact, we would argue that this is superior to using CNVs in this cancer type since the published evidence overwhelming support that G_q mutations occur before any CNVs and remain in 100% of tumors cells throughout tumor evolution (PMC5760704; PMC5619925). While sequencing and other technical issues can impact the calculation of G_q variant allele frequencies, the same is true for CNVs. To validate the use of G_q variant allele frequencies for purity estimation, we have analyzed the TCGA uveal melanoma data to correlate the G_q variant allele frequencies reported there with matching tumor purities inferred by TCGA copy number results. We found significant correlation between G_q mutation VAF-based tumor purity estimates with purity estimations established by copy number-based algorithms, FACETS (Pearson correlation = 0.4, $P < .0001$) and ABSOLUTE (Pearson correlation = 0.4, $P < .0001$). We thank the Reviewer for this idea, which further strengthens our claims. This supportive data has now been included in the manuscript (lines 292-294) and Supplementary Figure 7.

As requested, we now include tumor purities as a row in the oncprint in Figure 1a.

Whole genome doubling is rare in uveal melanoma and is generally limited to a small handful of large advanced cases, which comprised most cases in the TCGA but less than 10% in our study. Even within the large advanced enucleated cases of the TCGA, less than 7% of cases showed whole genome doubling (PMID 28810145). Thus, we do not believe that whole genome doubling is a concern in the interpretation of our results. We have added a comment acknowledging the possibility of whole genome doubling and why we do not believe this would alter our conclusions (lines 294-299).

Small tumors likely have lower purities because it is more likely that surrounding normal cells will be aspirated during tumor biopsy. We now explain this in the text (lines 239-241). We agree that one explanation for smaller tumors lacking a BSE mutation is that they could have been missed due to lower purities. However, since we detected a Gq mutation in all of these cases, a putative BSE mutation would have needed to be at low CCF for it to be missed, which still fits with our claim that BSE mutations arise later and are less likely to be present or detectable in small tumors. Thus, we believe that our method for estimating tumor purity is reasonable and does not negatively impact our inferences from these data. We now explain this in the text (lines 241-244).

My second concern is novelty. As the authors state, this sample set is one of the largest in uveal melanoma thus far. Yet the sample set itself is the subject of a previous publication, and almost all the findings here have been previously published, mostly by the authors themselves. I think the authors need to more convincingly delineate the novelty of this particular study, specifically which results are independent in this manuscript versus those that are simply extensions or validations of prior work.

RESPONSE: There are numerous examples of novelty in this manuscript. A few examples are listed here:

1. It is by far largest prospective study of NGS for uveal melanoma and the first to incorporate a large number of small tumors. As such, it provides the clearest evidence to date of the timing of genetic evolution in human samples spanning the entire spectrum from small to large tumors.
2. It is the first to compare uveal melanoma associated mutations to the standard-of-care 15-GEP/PRAME classifier in a large prospective dataset, showing convincingly that mutations are inferior to the 15-GEP/PRAME classifier for prognostication but have other value.
3. It is the first study to show the heretofore unrecognized importance of the SVM discriminant score as a biomarker for tumors in transition from Class 1 to Class 2, providing novel insights into the timing of genetic evolution. This will have important potential clinical value, especially in determining whether or not to treat small tumors on the border between benign and malignant features.
4. It provides the strongest evidence to date to address the controversy in the field as to whether GNA11 mutations are prognostically worse than GNAQ mutations, and whether they are independent risk factors for poor outcome.
5. It demonstrates the utility of the Gq mutation VAF to estimate tumor purity.

Here are my intermediate concerns:

Were the tumor samples collected prior to the treatments listed in Supplementary Table 2? Were these samples FFPE, frozen, or a mix?

RESPONSE: All biopsy specimens were collected and processed prior to treatment according to a standard operating procedure for clinical grade testing that was used consistently across all participating sites. This is clarified in the Methods section (line 339).

About 17% of tumor samples (n=290) failed sequencing, with the majority (n=234) having no UM driver mutation detected. Why? Providing some quality metrics of the sequencing results would help interpret this result.

RESPONSE: Since the NGS panel, which is not yet validated for clinical testing, was performed from the same biopsy specimen as that used for standard of care 15-GEP testing, we were ethically obligated to use the majority of the sample for the latter. This is the main explanation for the failure rate, which would likely be reduced dramatically in routine clinical use if the NGS become part of CAP-CLIA approved testing platform. Nevertheless, to satisfy the reviewer's concern, we have included library metrics in Supplementary Table 1, including sample DNA concentration, library concentration, number of reads, percentage of on-target reads, mean depth of coverage, and percentage of uniformity for targeted panel.

What is the evidence in this manuscript specifically that BAP1 loss causes changes to the tumor immune microenvironment that impact the transcriptomic shift from Class 1 to Class 2? This seems like a result from a prior publication focusing on cell line experiments, not genomic data of human tissues. The discussion on immune cells seems unrelated to the data here, yet it is a major part of Figure 5?

RESPONSE: The tumor immune microenvironment (TME) is pivotal to our central hypothesis as depicted in Figure 5 because several of the genes in the 15-GEP are expressed in infiltrating immune cells and contribute importantly to the overall GEP (PMID: 31980621). This is important for explaining why BAP1 loss is necessary but not sufficient for full conversion to Class 2, which requires subsequent reorganization of the TME. This explains why the transition from Class 1 to Class 2 is associated with a progressive, rather than immediate, inversion of the discriminant score.

The data availability is scant and the sequencing files should be provided in some form to a repository like dbGAP or EGA. The authors also need to provide sequencing coverage and mutant reads for each mutation so that readers can adequately assess the mutation calls.

RESPONSE: In addition to the data already made available to Reviewers, we will upload BAM files to dbGAP. We have also included sequencing and coverage metrics in Supplementary Table 1 (and in the public Dryad repository) for further assessment of mutation calls.

Minor concerns:

What was the approach for defining “small” as having thickness less than or equal to 2.5 mm and diameter less than or equal to 12 mm? It seems arbitrary based on the plot in Figure 2a.

RESPONSE: Please see response to similar question from Reviewer #2.

Figure 3 includes metastasis arrowheads in the legend but I do not see any on the timelines.

RESPONSE: Thank you for pointing this out. We have now corrected this in Figure 3.

The authors mention “punctuated” evolution in the abstract, which is a separate topic of their prior work. What is the evidence in this manuscript for “punctuated” evolution? All of the tumors here could have followed a step-wise paradigm before the sample was collected (even the “small” tumors were millions to billions of cells by the time of collection). For example, an inferred founder with multiple drivers is consistent with

“punctuated” evolution, but it is also consistent with step-wise evolution.

RESPONSE: The Reviewer makes a good point. We are happy to soften the statement to:

“Our findings suggest that the genetic aberrations that give rise to the archetypal evolutionary trajectories in UM usually start to arise early when tumors are small.” We have also removed similar language in the abstract.

Reviewer #4 (Remarks to the Author):

Thank you all for your time and thoughtful insights provided while reviewing our manuscript!